# The Potential Therapeutic Role of Metformin in Diabetic and Non-Diabetic Bone Impairment

**DOI:** 10.3390/ph15101274

**Published:** 2022-10-17

**Authors:** Wei Mu, Guoqiang Liang, Yue Feng, Yunyun Jiang, Falin Qu

**Affiliations:** 1Department of Pharmacy and Clinical Pharmacy, Precision Medicine Center, 904th Hospital of PLA, Wuxi 214044, China; 2Central Laboratory, Suzhou TCM Hospital Affiliated to Nanjing University of Chinese Medicine, Suzhou 215000, China

**Keywords:** metformin, AMPK, RUNX2, AGEs, osteogenesis

## Abstract

Metformin is a widely-used anti-diabetic drug in patients with type 2 diabetic mellitus (T2DM) due to its safety and efficacy in clinical. The classic effect of metformin on lowering blood glucose levels is to inhibit liver gluconeogenesis that reduces glucose production as well as increases peripheral glucose utilization. However, the factors such as hyperglycemia, insulin deficiency, reduced serum levels of insulin-like growth factor-1 (IGF-1) and osteocalcin, accumulation of advanced glycation end products (AGEs), especially in collagen, microangiopathy, and inflammation reduced bone quality in diabetic patients. However, hyperglycemia, insulin deficiency, reduced levels of insulin-like growth factor-1 (IGF-1) and osteocalcin in serum, accumulation of advanced glycation end products (AGEs) in collagen, microangiopathy, and inflammation, reduce bone quality in diabetic patients. Furthermore, the imbalance of AGE/RAGE results in bone fragility via attenuating osteogenesis. Thus, adequate glycemic control by medical intervention is necessary to prevent bone tissue alterations in diabetic patients. Metformin mainly activates adenosine 5′ -monophosphate-activated protein kinase (AMPK), and inhibits mitochondrial respiratory chain complex I in bone metabolism. In addition, metformin increases the expression of transcription factor runt-related transcription factor2 (RUNX2) and Sirtuin protein to regulate related gene expression in bone formation. Until now, there are a lot of preclinical or clinical findings on the application of metformin to promote bone repair. Taken together, metformin is considered as a potential medication for adjuvant therapy in bone metabolic disorders further to its antidiabetic effect. Taken together, as a conventional hypoglycemia drug with multifaceted effects, metformin has been considered a potential adjuvant drug for the treatment of bone metabolic disorders.

## 1. Introduction

Metformin is a widely prescribed oral antidiabetic drug for the treatment of type 2 diabetic mellitus (T2DM). Metformin is a member of the class of biguanide with the property of carrying two methyl substituents. The French scientist Tanret first extracted and isolated goat bean alkaloids (galegine) from *G. officinalis*, and conducted preliminary studies on this alkaloid in 1914. However, the hypoglycemic effect of guanidine was not discovered until 1918. Irish chemists Werner and Bell first prepared and obtained metformin in 1922. Unfortunately, metformin was first to be tried on humans for the treatment of diabetes by French diabetologist Jean Sterne until 1957. The advantage of low cost, safe profile and potent efficacy made metformin friendly and affordable for diabetic patients around the world for over 60 years. Metformin has become the first-line anti-diabetic medication without risk of hypoglycemia, comparing other anti-diabetic agents [1]. Previous studies show that the metformin stimulated adenosine 5′-monophosphate-activated protein kinase (AMPK) complex to regulate nutrition metabolism and inhibited mitochondrial respiratory chain I to sense intracellular energy state [2]. AMPK activation increases insulin sensitivity through sensitizing insulin receptor (IR) or insulin receptor substrate-1 (IRS1) allosterically, leading to more glucose transporter (GLUTs) transferring into the cell membrane that promotes extracellular glucose uptake [3,4]. Furthermore, metformin did not affect hormone secretion such as cortisol, glucagon, somatostatin, or growth hormone except glucagon-like peptide-1 (GLP-1) which is a potent stimulator of glucose-dependent insulin release from pancreatic b cells [5].

Currently, there is no available therapy for either type 1 or type 2 diabetes that completely restores normal function, so even in the best-responding patients, there is still a risk of many complications. New approaches to meet this high unmet medical need remain a top priority in metabolic disease research. Iminosugars, also known as an iminosaccharide, are any analog of sugar where a nitrogen atom has replaced the oxygen atom in the ring of the structure, have been evaluated for the potential treatment of diabetes as a consequence of their inherent glycosidase inhibition profiles [6]. Its proposed mode of action is through inhibition of intestinal a-1,4-glucosidases leading to a reduction in glucose absorption from the gut similar to the effect of acarbose [7,8,9,10]. Historically the attention paid to iminosugars lay in their powerful inhibition and modulation of carbohydrate processing enzymes, alterations which are implicated in a variety of diseases as the effects of metformin. As a result, the combined use of iminosugars and metformin might enhance the hypoglycemia effect without the side effects of hypoglycemia.

High risk of fractures usually occurred in type 1 diabetic mellitus (T1DM) patients with the property of low bone mineral density (BMD); however, it happened in T2DM patients with property of high body mass index (BMI) [11,12]. T2DM patients with a high risk of fracture and falling ascribed to a diabetic complication induced by polyol bypass, protein kinase C (PKC), hexosamine activation, increased advanced glycation end products (AGEs), and reactive oxygen species (ROS) production in mitochondria under hyperglycemia [13,14]. Another factor that neuropathy and chronic kidney disease caused by long-term hyperglycemia may be the reason for local bone mass loss increasing the risk of fracture. It is worth noting that some of the anti-diabetic medication taken by T2DM patients promoted skeletal fragility. Taking thiazolidinedione (TZD) for example, as an insulin sensitizer, it increased the risk of fracture in T2DM patients, particularly in women due to its mediation on precursor mesenchymal stem cells (MSCs) differentiation into the adipocyte lineage instead of osteoblast formation through activation of peroxisome proliferator-activated receptor gamma (PPARγ) [15,16]. Therefore, not all medications for glycemic control have underlying benefits for bone tissue improvement.

Metformin, a widely known antidiabetic medication, has an off-label effect on protecting damaged bone tissue in patients with T2DM [16]. Independent of metformin’s primary effect on hyperglycemia correction, it can directly promote progenitor cells differentiation into osteoblasts by activating the AMPK complex and increasing the levels of runt-related transcription factor2 (RUNX2) in related osteogenic gene expression [16]. In this review, the authors discuss the bone metabolic disorder under hyperglycemia and the mechanism of metformin in bone tissue improvement in the diabetic state. 

## 2. The Relationship between Hyperglycemia and Bone Impairment

### 2.1. Diabetic Induced the High Risk of Bone Fragility

Long-term hyperglycemia will result in glucose metabolism disorder, leading to adverse effects on microvascular function, glucose oxidative derivatives accumulation, and endocrine function impairment [17]. Both the bone formation from the differentiation of osteoblast and the bone resorption from the clearance of osteoclast heavily depend on energy sources from glucose metabolism [18]. The clinical evidence showed that DM patients had a high risk of osteoporosis fractures [19]. DM has been identified as one of the underlying high-risk factors in osteopenia and osteoporosis, both of which present an accumulation of AGEs in collagen. The abnormal accumulation of AGEs resulted in the reduction of normal collagen, which damaged osteoblastic differentiation and delayed fracture healing in DM patients [20]. Studies in vitro indicated that the increase of AGEs under hyperglycemia resulted in suppressing osteoblastic proliferation and promoting osteoclastic resorption [21,22]. Moreover, osteocalcin (OCN), only secreted by insulin-activated osteoblasts and a marker of bone mineralization, is regulated negatively under hyperglycemia [23]. Thus, hyperglycemia or insulin deficiency led to the accumulation of AGEs on the bone matrix and the reduction of OCN secretion, affecting the bone formation and mineralization on damaged bone tissue. Bone marrow progenitor cells (BMPCs), a common ancestor pluripotent cells of adipocytes and osteoblasts, were induced by RUNX2-mediated osteoblastic differentiation and PPARγ dominated adipocytic formation respectively [24]. Besides, the increase of ROS levels under hyperglycemia significantly accelerated the apoptosis of osteoblasts [25]. T1DM patients exhibited not only lower BMD but also the reduction of insulin-dependent osteo-anabolic because of damaged pancreatic β-cells, while, high bone resorption occurred in T2DM patients displayed higher bone resorption without BMD changes significantly [26,27]. Therefore, long-term hyperglycemia caused high risk of fracture or bone fragility, although the underlying mechanisms are also different in different types of diabetes models.

### 2.2. Key Role of AMPK Complex in Bone Metabolism

AMPK complex, a key energy sensor, played an important role in osteogenesis by regulating intracellular energy homeostasis and osteogenic-related hormone secretions [28,29]. Previous studies showed that the AMPK complex had three subunits (α, β, γ) encoded by seven genes forming AMPK heterotrimers [30]. In the complex, catalytic subunit α and regulatory β and γ subunits cooperate together to turn on catabolic pathways by sensing the ratio of AMP/ATP within cells [31]. AMPK complex not only regulated energy production to keep normal cell function, but is also involved in the upregulation of GLUT4 expression [32]. The activation of the AMPK complex inhibited the downstream mechanistic target of rapamycin complex (mTOR) to regulate cell cycle and growth [33]. Additionally, the AMPK activation stimulated MC3T3-E1 cells to differentiate into osteoblasts and 3T3-L1 cells into adipocytes through the regulation of the AMPK-Gfi1-OPN axis [34]. Likewise, the activation of AMPK stimulated the MSCs to differentiate into osteoblasts via the upregulation of RUNX2 and to differentiate into adipocytes by the mediation of PPARγ [35,36]. More interest, PPARγ increased the expression of receptor activator for nuclear factor-kB (RANKL) to stimulate osteoclastogenesis to promote bone resorption [30]. Furthermore, the phosphorylation of AMPKα can reduce the levels of intracellular PPARγ directly [37].

### 2.3. Insulin and Insulin-like Growth Factor-1 in Bone Formation

Previous studies showed that insulin or insulin-like growth factor-1 (IGF-1) had an important role in osteogenesis [38]. And the receptors of insulin and IGF-1 are widely distributed on the surface of osteogenic cells and displayed high activity in the differentiation of osteoblast [39,40]. In osteoblast, insulin or IGF-1 regulated the activity of RUNX2 by the classic Wnt/𝛽-catenin pathway and the induction of bone morphogenetic protein-2 (BMP-2) [39]. Furthermore, that insulin binding the receptors of osteoblast promoted the uptake of glucose for bone formation and binding the receptors of osteoclast increased the secretion of OCN regulating bone resorption and mineralization [26]. In a nutshell, the synthesis of extracellular collagen, the differentiation of osteoblast, and the induction of osteoclast depended on the regulation of insulin in vivo. Insulin deficiency in T1DM patients resulted in a low BMD due to osteoblast impairment, while T2DM patients without significant changes in BMD because of the damaged function of the osteoclast. IGF-1, a liver-specific protein factor, mainly acted on osteogenesis in a manner of the endocrine hormone. Hyperglycemia and AGEs accumulation in bone tissue matrix reduced the expression of IGF-1 receptors in osteoblast, thereby attenuating the response of osteoblast to IGF-1 even if the serum IGF-1 levels are normal [41] (Figure 1). More interest, there is a positive correlation between the levels of IGF-1 and OCN in serum [41]. Therefore, the reduction of serum IGF-1 or the deficiency of IGF-1 receptors on the surface of osteoblast led to bone fragility under hyperglycemia.

## 3. The Potential Mechanism of Metformin on Diabetic Bone Improvement

### 3.1. Activation of AMPKa and RUNX2 in Bone Formation

In the bone marrow progenitor cells, metformin targets on AMPK complex as a potent agonist [16]. In vitro studies, the phosphorylation of AMPKa promotes the differentiation and mineralization of osteoblasts [42]. RUNX2, as an osteoblast-specific transcription factor, plays an important role in osteocyte differentiation and bone formation [43]. Alkaline phosphatase (ALP), as an indicator of the early differentiation of osteoblasts, is closely associated with bone tissue remodeling and bone matrix mineralization [44]. The high levels of ALP expression indicate the ongoing differentiation of osteoblasts and the maturity of bone tissue [45]. In osteogenesis, the main effect of metformin is acting on AMPKa, upregulating the expression of RUNX2, and promoting the secretion of ALP [16].

Metformin inhibits AGE-induced inflammatory response in macrophages of mice through the phosphorylation of AMPKa and the suppression of the receptor of advanced glycation end products/nuclear factor-kB (RAGE/NF-kB) signal cascade [46,47] (Figure 2). The association between mTORC1 activation and Notch pathway was usually used to explain the impaired differentiation of preosteoblasts [48]. Previous studies showed that the main function of Notch signaling was to regulate the communication among neighboring cells and to determine their fates [49,50,51]. Besides, the activation of Notch signaling via the induction of signal transducer and activator of transcription3/p63/Jagged (STAT3/p63/Jagged) signaling cascade attenuated the differentiation of osteoblast too [52]. In addition, Metformin attenuated abnormal subchondral bone remodeling mediated by osteoclasts and alleviated early osteoarthritis through the AMPK/NF-kB/ERK signaling cascade without effect on blood glucose levels and body weight [53]. 

Metformin can stimulate bone formation at all glucose concentrations via increasing the expression of RUNX2 and upregulating the serum IGF-1 levels [54], and also significantly reducing apoptosis and the levels of intracellular ROS [55,56]. In an ovariectomized (OVX) rat model with bone cancer, metformin directly inhibited bone loss by inducing the expression of RUNX2 and low-density lipoprotein receptor-related proteins (LRP5) to upregulate the activity of bone marrow cells [57]. Notably, RUNX2 was regulated positively by OCN and osteoprotegrin (OPG), both of which are important osteoblast differentiation markers [58]. Moreover, mTORC1 regulated RUNX2 negatively via Notch signaling [59]. In addition, the phosphorylation of AMPKa increased the degradation of RUNX2 via the phosphorylation of the smad ubiquitination regulatory factor (SMURF) [18] (Figure 2). Metformin enhanced the differentiation of osteoblasts by the upregulation of RUNX2 via partner AMPK/upstream stimulatory factor-1(USF-1)/small heterodimer partner (SHP) axis [60] (Figure 2).

### 3.2. Upregulation of OPG/RANKL in Bone Resorption

Cortical and medullary bones were impaired in rats in a hyperglycemia state, which indicated that hyperglycemia delayed the osteogenic integration process [61,62]. In a DM rats’ model, the implant with metformin increased the levels of OPG rather than RANKL in the area of the implant, directly upregulating the ratio of OPG/RANKL [63]. It is clear that both OPG and RANKL are secreted by osteoblasts to activate osteoclast and regulate osteogenesis. Thus, the ratio of OPG to RANKL keeps in balance to maintain normal bone metabolism. It was reported that metformin upregulated the expression of OPG to prevent bone loss in estrogen deficiency rats with perispical lesions. [64,65] (Figure 2). Moreover, metformin partially reversed high tartrate-resistant acid phosphatase (TRAP) in osteoclasts, increased OCN secretion in osteoblasts, and reduced the ratio of OPG/RANKL in hyperglycemic animals with periodontitis [57].

### 3.3. Sirtuins in Bone Metabolism

A new hypothesis proposed that metformin enhanced the differentiation and proliferation of mouse preosteoblasts under a hyperglycemia state via SIRT6/NF-kB/OCT4 pathway [66] (Figure 2). Metformin promoted the expression of SIRT6 that deacetylates histone in nuclei to delay osteogenic-related gene transcription, inhibited NF-kB transport into the nucleus to reduce RANKL expression, and enhanced the expression of octamer-binding transcription factor 4 (OCT4) to stimulate cell differentiation in vitro [66]. Notably, dampened NF-kB in nuclei and the reduced expression of OCT4 were detected in sirt6 knockout (KO) mice [66]. Furthermore, metformin reversed H_2_O_2_-induced the apoptosis of osteoblast in OVX mice by upregulating NAD^+^ dependent protein deacetylase sirtuin-3 (SIRT3) expression via the PI3K/AKT pathway, which implied that metformin had a therapeutic effect in postmenopausal osteoporosis [67]. 

**Figure 2 pharmaceuticals-15-01274-f002:**
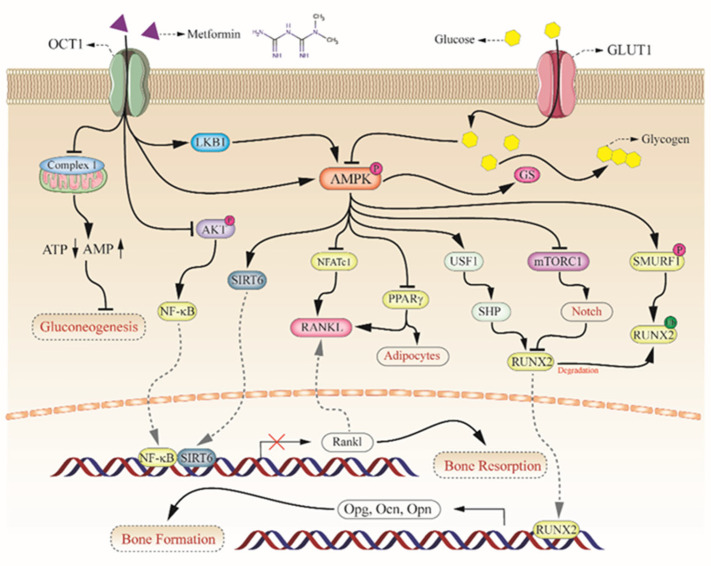
The effect of metformin on bone health. In the osteogenesis, metformin was pumped into osteogenic cells by OCT1, and inhibited gluconeogenesis through the inhibition of mitochondrial respiratory chain complex 1. Metformin inhibited the activation of AKT to reduce the production of cytoplasmic NF-kB, however, metformin activated AMPKa to enhance the expression of sirt6. In the nucleus, the binding of SIRT6 and NF-kB reduced the expression of RANKL which is a key ligand inducing the differentiation of osteoclast in bone remodeling. In addition, the activated AMPKa inhibits factors of NFATc1 and PPARg to repress the expression of RANKL respectively. However, it seems contradictory to the factor of RUNX2 in bone formation. Canonical AMPKa activation by metformin promoted the upregulation of RUNX2, however more extra glucose pumped into osteogenic cells by GLUT4 inhibited the phosphorylation of AMPKa resulting in the attenuation of downstream mTORC1 signaling cascade. It suggested that extra-high levels of glucose had the opposite effect in osteoblasts compared with that of metformin. Furthermore, AMPKa is a key regulatory factor in both in the generation and degradation of RUNX2. The activation of AMPKa promoted the activation of RUNX2 via the AMPK-USF1-SHP axis or mTORC1-Notch axis and mediated the degradation of RUNX2 via the AMPK-SMURF1 axis. Therefore, metformin, the extracellular levels of glucose, and the AMPK complex collaborated together to regulate bone formation and resorption. Refs. [16,18,46,47,60,61,62,63,64,65,66].

## 4. Effects of Metformin on Cells In Vitro

### 4.1. Effects on Stem Cells 

It is well known that the osteogenic potential of mesenchymal stem cells (MSCs) is seriously affected by persistent inflammation of periodontitis. Metformin carbon dots (MCDs) including citric acid and metformin hydrochloride effectively improved the activity of ALP, the formation of calcium deposition nodules, the expression of genes and osteogenic proteins in mesenchymal stem cells of the rat bone marrow (rBMSCs), thereafter effectively helped to regenerate lost alveolar bone in periodontitis rats [68]. A recent study showed that polycaprolactone/chitosan nanofibrous membranes containing metformin would be favored for bone regeneration as guided bone regeneration membranes because it was more suitable for cell proliferation, adhesion, and osteogenic differentiation of rBMSCs [69]. A study from canine BMSCs showed that metformin was a better osteogenic inducer for osteogenic differentiation in vitro [70]. Additionally, Metformin significantly induced osteogenic differentiation of human BMSCs while it was attenuated via inhibiting phosphorylation of glycogen synthase kinase 3b (GSK3b), which suggested that metformin had a potential effect on stimulating differentiation of human MSCs toward osteoblast [71].

### 4.2. Acts on Other Bone Cells

Metformin inhibited BMP6-induced Smad1/5 phosphorylation in osteoblast MC3T3-E1 cells, along with Smad6 up-regulation, and this effect was mitigated by the knockdown of Smad6 [72]. Thus, metformin may be a potential therapeutic drug for trauma-induced heterotopic ossification. A novel elastomeric biodegradable bone regenerative films developed from metformin and polyurethane (PU) exhibited that metformin present in PU formulation promoted adhesion, proliferation, and calcium deposition of MC3T3-E1 cell line [73]. In addition, metformin enhanced osteoblastic cell mobility in wound healing and migration assay and upregulated mark protein expression in osteoblastic differentiation in U2OS and MG63 cells while suppressing the differentiation of osteoclast in Raw 264.7 cells, and protected against ischemic necrosis in the rat femoral head epiphysis by preserving osteocyte function [74]. Metformin-treated preosteoblasts increased the expression of OPN which reduced the subsequent adherence of myeloma cells when they were silent. Proliferation markers were reduced in cocultured myeloma cells with preosteoblasts treated with metformin. Mice with 5TGM1 myeloma cells pre-treated with metformin had increased tumor loads, associated with increased osteolytic bone damage and high expression of the OPN in the bone marrow [75]. In an in vitro study of nondiabetic rats with a cranial defect model, metformin promoted the differentiation of rat adipose tissue (rASCs) into bone-forming cells. which osteogenic effect of metformin was also demonstrated with the rich calcium and phosphorous deposits on the newly formed mineralized extracellular matrix. [76].

## 5. Research on Bone Defect Animal Models

### 5.1. Promote Alveolar Bone Repair

In a critical-size alveolar bone defects model of rats, the Gelatin/nano-hydroxyapatite/metformin scaffold showed superior bone regeneration and promoted the synthesis of osteogenic proteins such as OCN, osteonectin, and collagen type I, which may be applied as a potential bone substitute to regenerate alveolar bone due to its good biocompatibility, interconnected pores allowing vascularization, relatively fast degradation, and higher bioactivity properties [77]. In a periodontitis rat model, local administration of chitosan-metformin based intra pocket dental film led to the reduction of alveolar bone destruction and displayed good antibacterial activity [78]. Additionally, the metformin-loaded b-tricalcium phosphate/Chitosan/mesoporous silica scaffolds implanted in the region of alveolar bone malformations in rats suffering from periodontitis promoted alveolar bone regeneration [79]. In another bone tissue engineering study, the group with metformin plus osteogenic had three- to four-fold increases over those of the osteogenic alone group in osteogenic gene expressions, ALP activity, and mineral synthesis, which demonstrated that human periodontal ligament stem cell (hPDLSCs) was a potent cell source for bone engineering and the calcium phosphate cement (CPC)-metformin scaffold with hPDLSCs was a highly promising construct to promote bone repair and regeneration effectively in craniofacial, dental, and orthopedic applications [80]. Thus, metformin might be an additional osteoinductive factor in osteogenesis.

### 5.2. Enhance Tendon-Bone Interface Healing

The healing of the tendon-bone interface (TBI) is a clinical dilemma that is closely related to the forming and remodeling of new bones at the repair site. A canine model study showed that the Achilles tendon-calcaneus (ATC) interfaces treated with metformin were repaired with a significantly higher fracture load and stiffness than the metformin-free test site. The micro-computed tomography (CT) analysis showed that the metformin-treated samples exhibited significantly higher bone volume/total volume and trabecular thickness than those of the metformin-free controls. These results were confirmed by hematoxylin and eosin (HE) staining as well. Immunohistochemical (IHC) staining showed that there were considerably more cells with OCN in newly formed bones with metformin-treated than those in the metformin-free control site at week 4. Furthermore, Masson’s trichrome staining showed that significantly more oriented collagen fibers anchored in the newly formed bone of the metformin-treated site than in the metformin-free control site [71]. Consequently, the local administration of metformin provided bone microarchitecture improvement at the calcaneus and an increase in the tensile properties of the repaired ATC interfaces in canines. These important findings demonstrated that the local administration of metformin may be an effective strategy for TBI healing in clinic.

### 5.3. Single Use of Metformin in Bone Repair

In a rat model of TiAl6Va4 implants on tibial bone, the ratio of peri implant bone tissue filling was higher in the metformin group than those in the control group, which suggested that systemic administration of metformin might increase titanium implant osseointegration in non-diabetic rats [81]. A poly lactic acid and polycaprolactone scaffold with the delivery of metformin-loaded gelatin nanocarriers enhanced the expression of the markers of osteogenic and angiogenic considerably and ameliorate bone in angiogenesis, growth, and defect reconstruction in a rat model of calvarial bone defects [82]. Additionally, metformin can accelerate bone healing and mature tissue formation at a fracture site in a cranial defect rats’ model [83]. In collagen-induced arthritis (CIA) model rats, metformin significantly inhibited systemic inflammation and synovitis, the changes of trabecular bone and degradation of the cartilage layer matrix, and osteoclast formation in the knee joint, and the apoptosis of chondrocytes [83]. In a chronic kidney disease-mineral and bone disorder (CKD-MBD) rat model, metformin protected against the development toward severe CKD to prevent vascular calcification development and high bone turnover disease progression, but there was no evidence of the reduction of aorta or small vessel calcification [84]. 

Moreover, metformin effectively increased the levels of serum ALP in the ketogenic diet (KD) mice while reducing the levels of serum TRAP in the OVX mice, but the OCN expression up-regulated and the TRAP expression down-regulated in both OVX and KD mice [85]. This study revealed that metformin can effectively alleviate KD-induced cancellous bone loss and maintain the biomechanical properties of long bones, which suggested that metformin was a potential drug for the treatment of KD-induced osteoporosis in teenage [85]. In ultra-high-molecular-weight polyethylene particle-induced osteolysis mouse models, metformin reduced dickkopf-related protein 1 (DKK1), and sclerostin that is the negative regulator of bone formation, and increased OPG secretion and the ratio of OPG/RANKL to exert the property of bone protect [86]. These findings suggested that metformin-induced differentiation and mineralization of osteoblasts, while it inhibited osteoclastogenesis through the secretion of mature osteocytes [86]. A systematic review was conducted in accordance with the 2020 PRISMA guidelines to evaluate the evidence supporting the bone-protective effects of metformin on male animal models with T2DM.

This study shows that metformin enhanced bone density and reduced the effects of T2DM on fat formation in animal models, however, further research is needed to determine the optimal dose of metformin needed to show these therapeutic effects [87].

### 5.4. Combinational Use of Metformin in Bone Repair

A novel poly L-lactic acid/nanoscale hydroxyapatite/metformin nanocomposite scaffold had the dual function of tumor repression and bone repair, which provides a promising new therapy for tumor-induced bone defects [88]. The metformin-incorporated nano-gelatin/hydroxyapatite fibers upregulated osteogenic gene and protein expression, and greatly improved healing potential in a rat model of forearm critical bone defect [89]. A study on the combined use of metformin and alendronate showed that the alendronate use alone can increase serum GLP-1 levels significantly and the use of metformin alone can improve bone microstructure like Tb.Sp and Tb.N of the spinal in the control group. Consequently, metformin and alendronate in combination can improve the progress of glucose metabolism and bone metabolism such as lowering blood glucose levels, increasing glucose tolerance, increasing insulin sensitivity, and reducing bone loss than the control group, however, they do not appear to act in a clearly synergistic manner in their combined use [90].

## 6. Investigations on Clinical Settings

A large cohort study of Chinese patients with T2DM including 11,458 T2DM patients aged no less than 40 years showed that the overall prevalence of osteopenia and osteoporosis was 37.4% and 10.3% respectively, and was lower in metformin-treated patients (34.6% vs. 38.3% and 7.1% vs. 11.3%, both *p* < 0.001) [91]. Patients who had older age, a lower BMI, and estimated glomerular filtration rate (eGFR), had more osteoporosis, a lower BMD and T-score at the femoral neck (FN), lumbar spine (LS), and total hip (TH) [84]. Metformin use and the male sex were associated with a higher BMD. Metformin treatment was also independently associated with higher T-score at LS, FN and TH (b = 0.120, 0.082 and 0.108; all *p* < 0.001), and lower odds ratio (OR) of osteoporosis (OR = 0.779, 95%CI: 0.648–0.937, *p* = 0.008) or low BMD (OR = 0.834, 95%CI: 0.752–0.925, *p* = 0.001) [91]. However, when analyzed by sex, this association of a lower OR for osteoporosis with metformin was only significant in women. (OR = 0.775, 95%CI:0.633–0.948; *p* = 0.013) [91]. Consequently, metformin treatment was associated with a lower OR of osteoporosis and a higher T-score, particularly in the female population, regardless of age, BMI, and eGFR [91].

Currently, a prospective study with enrollments of 142 patients with T2DM treated with metformin or metformin plus a-glucosidase inhibitors was conducted in China. Their results showed that patients with metformin plus a-glucosidase inhibitors were associated with significantly lower levels of 2-h postprandial blood glucose (2hPG), hemoglobin A1c (HbA1c), fasting plasma glucose (FPG), and homeostasis model assessment-insulin resistance (HOMA-IR) vs. metformin alone (*p* < 0.05) after 12 weeks treatment [92]. The BMD index was correlated with IGF-1R positively and with vascular endothelial growth factor (VEGF) and endothelin negatively after treatment in both groups [92]. Metformin plus a-glucosidase inhibitors can effectively control blood glucose and reduce HOMA-IR in patients with primary T2DM, however, a large sample study was essential to predict osteoporosis development in T2DM patients [92]. 

A study about the effect of metformin on primary bone cancer risk conducted by Taiwan’s National Health Insurance showed that the incidence rates were 10.56 and 12.90 per 100,000 person-years for 453,532 metformin initiators and 220,000 non-metformin initiators respectively, and the hazard ratio between initiators and non-initiators was 0.830 (*p* = 0.0551) in the intention-to-treat analysis. Additionally, the incidence rates were 7.58 and 11.77 per 100,000 person-years, respectively, and the risk ratio was 0.615 (*p* = 0.0005) in the per-protocol analysis [93]. In addition, metformin treatment in patients with excess endogenous glucocorticoid showed potential protective effects by reducing bone resorption, thereby reducing the undesirable side effects of glucocorticoid treatment [94]. 

In addition, one cannot ignore the side effects of metformin when it is applied in clinical settings. The most common adverse effect of metformin is gastrointestinal irritation, including diarrhea, cramps, nausea, vomiting, and increased flatulence; metformin is more commonly associated with gastrointestinal adverse effects than most other antidiabetic medications [95]. The most serious potential adverse effect of metformin is lactic acidosis; this complication is rare, and the vast majority of these cases seem to be related to conditions such as impaired liver or kidney function, rather than to the metformin itself [96]. Metformin is not approved for use in those with severe kidney disease, but may still be used at lower doses in those with kidney problems [97]. Lactic acidosis almost never occurs with metformin exposure during routine medical care [98]. Rates of metformin-associated lactic acidosis are about nine per 100,000 persons/year, which is similar to the background rate of lactic acidosis in the general population [99]. A systematic review concluded no data exists to definitively link metformin to lactic acidosis [100]. The risk of metformin-associated lactic acidosis is also increased by a massive overdose of metformin, although even quite large doses are often not fatal [101].

## 7. Discussion and Perspective

Diabetic patients are at high risk of bone and joint problems, such as osteoporosis and bone fractures. It is well known that metformin can reduce blood glucose levels by inhibiting hepatic gluconeogenesis, decreasing the intestinal absorption of glucose, as well as increasing insulin sensitivity by promoting peripheral glucose uptake and utilization. It is well established that metformin inhibits mitochondrial complex I activity, which contributed to the potent antidiabetic effects of metformin [102,103]. The mitochondrial complex I inhibited by metformin led to the reduction of ATP production, and subsequently inhibit fructose-1,6-bisphosphatase (F-1,6-BP) enzyme to attenuate cell gluconeogenesis [104,105,106]. Furthermore, the activation of AMPKa by metformin phosphorylated both two isoforms of acetyl-CoA carboxylase (ACC) enzyme, thereafter inhibiting the fat synthesis and stimulating fat oxidation that reduced hepatic lipid stores and increased liver sensitivity to insulin [104]. In the gut, metformin increased anaerobic glucose metabolism in enterocytes to reduce the net glucose uptake and promoted GLP-1 secretion from small intestinal endometrial cells to increase the glucose utilization of the gut [104]. 

Treatment with metformin not only provided adequate glycemic control, but also alleviated diabetic complications and reduced the risk of bone fracture. For patients with T1DM, the changes in BMD provided a good diagnostic mark in bone fragility induced by hyperglycemia. However, it was difficult in diagnosing bone impairment for patients with T2DM in the clinic with regard to BMD index, because the BMD index in T2DM patients keeps normal or even higher than that of the nondiabetic. Currently, there is no suitable method to assess the degree of bone fragility for patients with T2DM. Thus, to deal with bone impairment under hyperglycemia, it is important to take medicine on glycemic control to prevent fracture for diabetic patients except for TZDs treatment that had a side effect on the aggravation of bone fragility through upregulation of PPARγ. Notably, both high and low doses of metformin use can improve bone impairment in diabetic patients, especially for patients with T2DM [107]. 

The accumulation of AGEs and upregulation of the expression of RAGEs are common in diabetic patients because the cells regulate their own stress state in response to the changes in the environment under hyperglycemia [107,108,109,110]. The binding of insulin and its receptors on the surface of osteoblasts stimulated the secretion of OCN, which in turn promoted the proliferation of β-cells in the pancreas and insulin sensitivity, however, hyperglycemia and high levels of ROS regulated the production of OCN negatively [111]. Moreover, the activation of AMPKa by metformin inhibited intracellular levels of nuclear factor of activated T-cell cytoplasmic 1 (NFATc1), and the activity of 3-hydroxy-3-methylglutaryl-coenzyme A (HMG-CoA) reductase in the mevalonate pathway, which exerted a positive influence on bone tissue [112,113,114]. Previous studies showed that metformin had a potentially positive effect on 20% reductions of fracture risk in T2DM patients [115]. In addition, metformin is indirectly involved in the action of ERK activation and the induction of inducible nitric oxide synthase (iNOS), both of which are an important regulative factor in osteogenesis [68]. Since metformin has no substantial effect or side effect on the control of glucose levels in nondiabetic individuals, it may be regarded as a potential adjuvant therapeutic drug for bone disorders in patients without diabetes. 

In addition to the mechanisms and application of metformin in bone improvement described above, more investigations are needed on the complex pathophysiology of osteoporosis induced by hyperglycemia and the effect of metformin in bone quality improvement. Recently, the bone fragility in the diabetic in Europe-towards a personalized medicine approach (FIDELIO) consortium has been founded. This research network will supply a platform for the application of next-generation techniques on the aspect of genetic epidemiological and biological pathways of diabetic bone disease via Mendelian randomization. For another, head-to-head studies are needed to compare metformin with other anti-osteoporotic drugs for their different efficacy in fracture prevention in the diabetic or nondiabetic. Last but not the least, individual differences such as heredity, gender, age, height, weight, lifestyle, and disease states, and external factor such as drug to drug interactions (DDI) should not be ignored in the assessment of metformin in osteoporosis prevention. It is essential to put pharmacogenetic studies on the agenda to take genetic polymorphisms including drug metabolic enzymes, transporters, and receptors into consideration for the better efficacy difference assessment of anti-osteoporosis agents.

In this review, the authors discussed the link between hyperglycemia and bone fragility, and the relationship between metformin and relevant signaling pathways involved in bone metabolism. A lot of preclinical findings have shown that metformin is a potent chemical agent in the promotion of osteogenic formation in diabetic or nondiabetic animal models. Furthermore, multiple clinical trials on the bone protection effect of metformin have been conducted to reduce bone fragility in diabetic or nondiabetic patients throughout the world (clinicaltrials.gov, accessed on 16 March 2022). All in all, metformin may be a promising candidate drug for the treatment of diabetic or nondiabetic bone impairment in the future.

## Figures and Tables

**Figure 1 pharmaceuticals-15-01274-f001:**
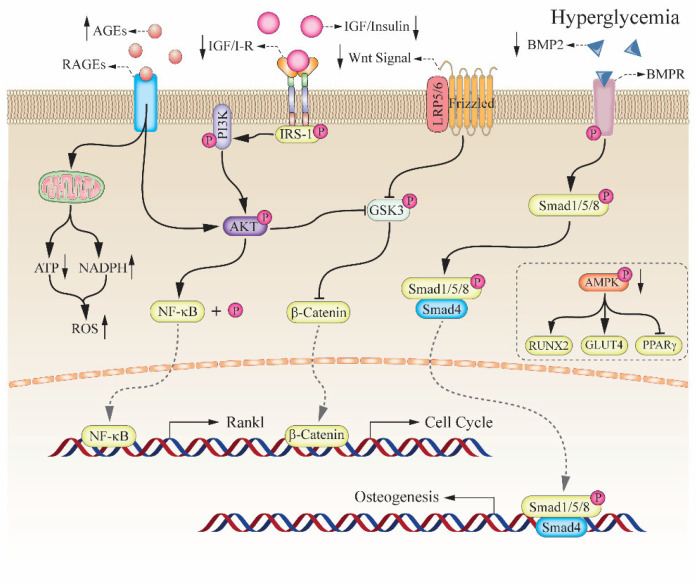
Bone fragility induced by hyperglycemia. High levels of AGEs induced by hyperglycemia resulted in the increase of NADPH and ATP, which lead to high levels of ROS in osteoblastic cells. Insulin deficiency and desensitization of insulin receptors reduced phosphorylation of PI3K through the action of insulin binding its receptor the same as the IGF-1 pathway. Extracellular AGEs binding RAGEs or PI3K stimulate the activation of AKT that thereafter inhibited the phosphorylation of GSK3b that suppresses b-catenin accumulation in the downstream of Wnt signal pathway and dephosphorylates NF-kB that promotes the expression of the RANKL in osteoblast. Furthermore, the downregulation of BMP2 induced by hyperglycemia induces dampened the binding between Smad1/5/8 and Samd4 in the cytoplasm, which blocked that process of osteogenesis. In addition, that the reduced activity of AMPKa under hyperglycemia dismissed the inhibition of PPARg and decreased the expression of RUNX2 and GLUT4, which affected the differentiation of the MSCs into osteoblasts. Refs. [20,21,22,23,24,32,37,39,41].

## Data Availability

Not applicable.

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
