# Peer review of "The Potential Therapeutic Role of Metformin in Diabetic and Non-Diabetic Bone Impairment"

_pharmaceuticals, 2022, doi:10.3390/ph15101274_

Round 1

Reviewer 1 Report

In this review article the author presented the potential therapeutic role of metformin in diabetic as well as non-diabetic bone impairment. They presented all studies/results on metformin to promote bone repair reported in literature and it will be regarded as an important review on this topic. My comments are presented below.

1. The title of the article is presented appropriately. The language used in the manuscript is rational and flawless.

2. The abstract is very concise and reflects the objective of the manuscript. No typos are observed.

3. The introduction section is also presented appropriately.

(It is advisable to present the chemical structure of metformin with a brief information of its innovator)

4. In Fig:1, reference should be mentioned.

5. In section 2, the relation between relationship between hyperglycemia and bone impairment is thoroughly discussed with proper references.

6. In Fig:2, reference should be mentioned.

7. Schematic representation for section 3.2 (Upregulation of OPG/RANKL in bone resorption) and section 3.3 (Sirtuins in bone metabolism) may be incorpated like section 3.1.

8. In section 4: Effects of metformin on cells in vitro, the authors have briefly discussed with recent literatures. (in each sub sections it is advisable to present some literature elaborately with couple of schemes)

9. In section 6 (Investigations on clinical settings), it is advisable to elaborate. Please mention is there any adverse side effects of metformin.

10. The authors are recommended to emphasis the importance of iminosugars and sugar derivatives as an anti-diabetic agent, and it is recommended to cite following relevant articles related to iminosugars in the introduction section.

  • Nash, R. J.; Kato, A.; Yu, C-. Y.; Fleet, G. W. J. Iminosugars as therapeutic agents: recent advances and promising trends. Future Med. Chem. 2011, 3, 1513−1521.
  • Yang, L.-F.; Shimadate, Y.; Kato, A.; Li, Y.-X.; Jia, Y.-M.; Fleet, G.W.J.; Yu, C.-Y. Synthesis and glycosidase inhibition of N-substituted derivatives of DIM. Org. Biomol. Chem. 2020, 18, 999–1011.
  • Chennaiah, A.; Dahiya, A.; Dubbu, S.; Vankar, Y. D. A Stereoselective Synthesis of an Imino Glycal: Application in the Synthesis of (−)-1-Epi -Adenophorine and a Homoiminosugar. Eur. J. Org. Chem. 2018, 6574−6581.
  • Chennaiah, A.; Bhowmick, S.; Vankar, Y. D. Conversion of glycals into vicinal-1,2-diazides and 1,2-(or 2,1)-azidoacetates using hypervalent iodine reagents and Me3SiN3. Application in the synthesis of N-glycopeptides, pseudo-trisaccharides and an iminosugar. RSC Adv. 2017, 7, 41755−41762.
  • Rajasekaran, P.; Ande, C.; Vankar, Y. D. Synthesis of (5,6 & 6,6)-oxa-oxa annulated sugars as glycosidase inhibitors from 2-formyl galactal using iodocyclization as a key step. ARKIVOC 2022, vi, 5−23.

11. The authors are suggested to cite the following relevant articles related to metformin.

i. Pharmaceuticals 2020, 13(9), 234; https://doi.org/10.3390/ph13090234

ii. Pharmaceuticals 2021, 14(2), 122; https://doi.org/10.3390/ph14020122

 Overall, after addressing the points mentioned above, I recommend this review to publish in pharmaceuticals.

Author Response

  1. Open Review

Comments and Suggestions for Authors

In this review article the author presented the potential therapeutic role of metformin in diabetic as well as non-diabetic bone impairment. They presented all studies/results on metformin to promote bone repair reported in literature and it will be regarded as an important review on this topic. My comments are presented below.

  1. The title of the article is presented appropriately. The language used in the manuscript is rational and flawless.

Response: Thanks for your positive comments and evaluation on the title of our manuscript.

  1. The abstract is very concise and reflects the objective of the manuscript. No typos are observed.

Response: Thanks for your positive comments and evaluation on the abstract section of our manuscript.

  1. The introduction section is also presented appropriately.

Response: Thanks for your positive comments and evaluation on the introduction section of our manuscript. We added the chemical structure of metformin with a brief information of its innovator, marked with red in revised manuscript.

(It is advisable to present the chemical structure of metformin with a brief information of its innovator)

  1. In Fig:1, reference should be mentioned.

Response: relevant references were added in Fig.1 description with red mark in revised manuscript. Refs(20-24, 32, 37, 39, 41).

  1. In section 2, the relation between relationship between hyperglycemia and bone impairment is thoroughly discussed with proper references.

Response: Thanks for your positive comments and evaluation on the section 2 of our manuscript.

  1. In Fig:2, reference should be mentioned.

Response: relevant references were added in Fig.2 description with red mark in revised manuscript. Refs(16, 18, 46, 47, 60, 65, 66)

  1. Schematic representation for section 3.2(Upregulation of OPG/RANKL in bone resorption) and section 3.3(Sirtuins in bone metabolism) may be incorpated like section 3.1.

Response: It’s a good suggestion for our manuscript. The interaction between OPG and RANKL in bone resorption depends on the change of ratio, which is difficulty displayed in schematic representation. For sirtuin in section 3.3, the interaction of sirt6 and NF-kB has been exhibited in schematic representation, however the function of oct4 gene is about the process of development as a role of transcription factor. Thus, almost all the elements of section 3.1, 3.2 and 3.3 displayed in Fig. 2.

  1. In section 4: Effects of metformin on cells in vitro, the authors have briefly discussed with recent literatures. (in each sub sections it is advisable to present some literature elaborately with couple of schemes)

Response: Thanks for your advice. All the literatures we selected to describe the effects of metformin in each section are important equally. So, we use 1-2 sentences to describe its importance to underline the effect of metformin. Furthermore, we only have 10 days for manuscript revision, it’s not enough for us to present literature elaborately in each sub sections. If the time is sufficient, we can revise each section elaborately. 

  1. In section 6 (Investigations on clinical settings), it is advisable to elaborate. Please mention is there any adverse side effects of metformin.

Response: thanks for your advice, in section 6 of our manuscript, we have elaborately described the Refs (91, 92, 93) in three separate paragraphs to present the information on the positive results of clinical use of metformin. In this section, we added adverse side effects of metformin in revised manuscript in red mark. Refs (95-101) are marked in red.

  1. The authors are recommended to emphasis the importance of iminosugars and sugar derivatives as an anti-diabetic agent, and it is recommended to cite following relevant articles related to iminosugars in the introduction section.
  • Nash, R. J.; Kato, A.; Yu, C. Y.; Fleet, G. W. J. Iminosugars as therapeutic agents: recent advances and promising trends. Future Med. Chem20113, 1513−1521.
  • Yang, L.F.; Shimadate, Y.; Kato, A.; Li, Y.-X.; Jia, Y.-M.; Fleet, G.W.J.; Yu, C.-Y. Synthesis and glycosidase inhibition of N-substituted derivatives of DIM. Org. Biomol. Chem2020, 18, 999–1011.
  • Chennaiah, A.; Dahiya, A.; Dubbu, S.; Vankar, Y. D. A Stereoselective Synthesis of an Imino Glycal: Application in the Synthesis of (−)-1-Epi -Adenophorine and a Homoiminosugar. Eur. J. Org. Chem2018, 6574−6581.
  • Chennaiah, A.; Bhowmick, S.; Vankar, Y. D. Conversion of glycals into vicinal-1,2-diazides and 1,2-(or 2,1)-azidoacetates using hypervalent iodine reagents and Me3SiN3. Application in the synthesis of N-glycopeptides, pseudo-trisaccharides and an iminosugar. RSC Adv20177, 41755−41762.
  • Rajasekaran, P.; Ande, C.; Vankar, Y. D. Synthesis of (5,6 & 6,6)-oxa-oxa annulated sugars as glycosidase inhibitors from 2-formyl galactal using iodocyclization as a key step. ARKIVOC 2022, vi, 5−23.

Response: It’s a good idea for us to revise manuscript, In the introduction section, we added content of iminosugar as an anti-diabetic agent and cited relevant articles related to iminosugars in the second paragraph of the introduction section. We marked in red in the introduction of revised manuscript. Refs (6-10) are marked in red.

11. The authors are suggested to cite the following relevant articles related to metformin.                   

Pharmaceuticals 2020, 13(9), 234; https://doi.org/10.3390/ph13090234

Pharmaceuticals 2021, 14(2), 122; https://doi.org/10.3390/ph14020122

 Overall, after addressing the points mentioned above, I recommend this review to publish in pharmaceuticals.

Response: thanks for your advice, we added these two papers about metformin mentioned above in revised manuscript and marked in red in the section of reference. Refs (14, 47) are marked in red.

14. Drzewoski, J.; Hanefeld, M., The Current and Potential Therapeutic Use of Metformin-The Good Old Drug. Pharmaceuticals (Basel) 2022, 14, 122

47. Salvatore, T.; Pafundi, P.C.; Galiero, R.; Gjeloshi, K.; Masini, F.; Acierno, C.; Di Martino, A.; Albanese, G.; Alfano, M.; Rinaldi, L.; Sasso, F.C., Metformin: A Potential Therapeutic Tool for Rheumatologists. Pharmaceuticals (Basel) 2020, 13, 234.

Reviewer 2 Report

The review article “The potential therapeutic role of metformin in diabetic and non-diabetic bone impairment” by Wei Mu et al. attempts to provide a reasonable information about role of metformin in diabetic and 2 non-diabetic bone impairment. Following changes are required before its considered for publication:

Abstract:

1. The sentence “However, the factors such as hyperglycemia, ……………….reduced bone quality in diabetic patients” needs to be re-written for sentence structure.

2. Similarly, this sentence also requires re-write up “Taken together, ……… further to its antidiabetic effect”

Introduction:

1. First and last paragraph of introduction gives information about metformin. Authors should explain the difference between these two different paragraphs giving similar kind of information. Authors should consider merging into single paragraph

2. Authors should generally discuss the mode of action of metformin in this section and what property of metformin could be responsible for bone features. A paragraph linking them in introduction section will be important.

General comments:

1. Authors should make a table that represent the author, year, type/number of patients, dose, concomitant disease (if any) and metformin end result on bone features.

2. Whole manuscript should be checked for English language improvement. Correction by native English speaker is preferred.

3. Check italics. Most of the words that should be italic are not considered. For example, In-vitro, in-vivo, via etc.

4. Authors should attach the similarity report from “ithenticate” preferably. Similarity should not be more than 10%

Author Response

2.Open Review

Comments and Suggestions for Authors

The review article “The potential therapeutic role of metformin in diabetic and non-diabetic bone impairment” by Wei Mu et al. attempts to provide a reasonable information about role of metformin in diabetic and non-diabetic bone impairment. Following changes are required before its considered for publication:

Abstract:

  1. The sentence “However, the factors such as hyperglycemia, ……………….reduced bone quality in diabetic patients” needs to be re-written for sentence structure.

Response: thanks for your advice, we rewrite this sentence structure as “However, hyperglycemia, insulin deficiency, reduced levels of insulin-like growth factor-1 (IGF-1) and osteocalcin in serum, accumulation of advanced glycation end products (AGEs) in collagen, microangiopathy, and inflammation, reduce bone quality in diabetic patients.” in abstract section in revised manuscript and marked in purple.

  1. Similarly, this sentence also requires re-write up “Taken together, ……… further to its antidiabetic effect”

Response: it’s a good suggestion with regard to this sentence, we rewrite this sentence as “Taken together, as a conventional hypoglycemia drug with multifaceted effects, metformin has been considered as a potential adjuvant drug for the treatment of bone metabolic disorders.” in abstract section in revised manuscript and mark in purple.

Introduction:

  1. First and last paragraph of introduction gives information about metformin. Authors should explain the difference between these two different paragraphs giving similar kind of information. Authors should consider merging into single paragraph

Response: Thanks for your advice. In the introduction section of our manuscript, the first paragraph we described the general effect of metformin, the second one is the description on the relationship between hyperglycemia and bone fragility, and the third one is about the potential effect of metformin on bone repairment. In the context, we use this paragraph structure to better introduce the following description of metformin. If it has to be merged into one paragraph, we will reconstruct it in terms of the context. In our opinion, we believed this structure was proper in our manuscript.

  1. Authors should generally discuss the mode of action of metformin in this section and what property of metformin could be responsible for bone features. A paragraph linking them in introduction section will be important.

Response: it’s a good idea for revision, thanks for your advice again. In the introduction section, we have described the general effect of metformin in hypoglycemia and the reason that hyperglycemia resulted in bone fragility. For the detail information, we showed it in the following paragraphs (section2 and section3) in our manuscript. Currently, there is no exact and consistent conclusion on the effect of metformin on bone fragility improvement in the diabetic or nondiabetic. It is not sufficient to draw consistent conclusions with regard to current literatures. Thus, we should do more research on the mechanism of metformin acts on bone fracture, although there are some clinical experiments conducted by FDA (clinicaltrials.gov).

General comments:

  1. Authors should make a table that represent the author, year, type/number of patients, dose, concomitant disease (if any) and metformin end result on bone features.

Response: it’s a good suggestion, however, the information about clinical results of metformin on bone diseases is not sufficient to draw convincible conclusions. Currently, most of the research on the metformin acting on bone are in the laboratory stage. There are few clinical phase trials from the official website of FDA (clinicaltrials.gov). In the section of Investigations on clinical settings of our manuscript, we use 3 proper papers to describe the recent clinical research on the bone disease with metformin treatment. As a result, due to literature deficiency, the table you advised can’t be made based on current available clinical investigations. We hope that you understand its difficulty. 

  1. Whole manuscript should be checked for English language improvement. Correction by native English speaker is preferred.

Response: thanks for your advice, we have English language improvement by Elsevier Language Editing Service, because we submitted our manuscript to one journal under Elsevier at the first time. So, we chose Elsevier Language Editing Service for language improvement before submission. In addition, the Pharmaceutical Journal can supply language editing service before manuscript publishing, there is no problem on the issue of language.

  1. Check italics. Most of the words that should be italic are not considered. For example, In-vitro, in-vivo, viaetc.

Response: thanks for your reminding, we checked italics and marked them in purple in our revised manuscript.

  1. Authors should attach the similarity report from “ithenticate” preferably. Similarity should not be more than 10%

Response: thanks for your suggestion, the assistant editor of Pharmaceutical has checked the similarity our manuscript before reviewing. So, our manuscript reached the similarity standard of the Pharmaceutical Journal, although we don’t know the final results of text similarity. If you request that the similarity of our manuscript is not be more than 10%, we can check it again by “ithenticate” for further consideration. However, the academic editor gave us only 10 days for major revision of our manuscript, there is not enough time for us to do it if the similarity of our manuscript were more than 10%. If we have more time to do it, we would like to do such work. We hope you can understand our situation. Thanks a lot.